# Feeding Behaviour and Lifestyle of Children and Adolescents One Year after Lockdown by the COVID-19 Pandemic in Chile

**DOI:** 10.3390/nu13114138

**Published:** 2021-11-19

**Authors:** Edson Bustos-Arriagada, Sergio Fuentealba-Urra, Karina Etchegaray-Armijo, Nicolás Quintana-Aguirre, Oscar Castillo-Valenzuela

**Affiliations:** 1School of Nutrition and Dietetics, Faculty of Medicine, Universidad Finis Terrae, Santiago 7501015, Chile; ketchegaraya@uft.edu (K.E.-A.); nquintanaa@uft.edu (N.Q.-A.); ocastillo@uft.cl (O.C.-V.); 2Escuela de Educación, Facultad de Educación y Ciencias Sociales, Universidad Andres Bello, Concepción 4030000, Chile; sergio.fuentealba@unab.cl

**Keywords:** COVID-19, lifestyle, feeding behaviour, children, adolescents

## Abstract

Lockdown caused by the COVID-19 pandemic may have influenced feeding behaviour and lifestyle in children and adolescents. The purpose of this study was to analyse feeding behaviour and lifestyle in children and adolescents one year after lockdown by the COVID-19 pandemic in Chile. In this cross-sectional study an online survey was implemented in 1083 parents and caregivers regarding their children’s feeding behaviour and lifestyle and sociodemographic background. The results showed that “eat breakfast daily” (89.2%), “not overnight food intake” (69.9%) and “not fast-food intake” (66.0%) were the most frequent reported feeding behaviours, particularly in pre-school children. Respondents declaring healthy feeding behaviours and lifestyle were 23.4 and 23.7%, respectively, with no significant differences by sex. In pre-school children, families with three or fewer members and parents or caregivers with an undergraduate or postgraduate degree reported a significantly better feeding behaviour and lifestyle compared to families with more than three members and parents or caregivers without an undergraduate or postgraduate degree. In conclusion, the pandemic lockdown had a negative impact in lifestyle in children and particularly in adolescents. Healthier feeding behaviour was associated with fewer family members and parents or caregivers with at least an undergraduate degree.

## 1. Introduction

In December 2019, a pneumonia outbreak of unknown aetiology emerged in the city of Wuhan, China, alerting medical and scientific communities around the world [1,2,3]. In March 2020, the respiratory illness caused by the SARS-CoV-2 virus (COVID-19) was declared a pandemic by the World Health Organization (WHO). Since then, the disease has killed millions of people around the world [4,5]. In Chile and other countries, the main action to prevent the spread of the virus was a total lockdown of the population in 2020 [5]. This restriction caused different alterations in the behaviour and daily living activities of Chilean families [4].

The lockdown has influenced lifestyles of children and young people, particularly regarding food intake habits, levels of physical activity, leisure and sleeping time [6]. The WHO indicates that healthy diet and lifestyle help to prevent and treat COVID-19 [7]. During childhood and adolescence, nutrition is crucial for growth. In addition, habits of physical activity and adequate sleeping hours are key factors determining well-being and quality of life that consequently may prevent chronic non-communicable diseases [8,9]. According to the National Board for School Assistance and Scholarships (JUNAEB, Chile) overnutrition increased 54% in children and adolescents in 2020 compared to 2019. This increment may be explained by the lockdown during the COVID-19 pandemic [10]. Studies have revealed an increase in the number of daily meals in young children [11]. Similarly, adolescents have shown worse results in weight management lifestyle programs while at home, compared to when they were at school [12,13].

Movement restrictions and lockdown can produce food shortages and limit fresh food availability. As recognized by the Food and Agriculture Organization of the United Nations (FAO), the COVID-19 pandemic has interrupted the food chain around the world, and people with lower economical resources are mainly affected by this limited fresh food access. This may have increased the consumption of inappropriate foods in people with lower incomes [9,14].

Therefore, the purpose of this research was to examine the influence of the COVID-19 lockdown on the feeding behaviour and lifestyle of Chilean children and adolescents and to compare this information with international recommendations and local dietary guidelines regarding balanced diet, ultra-processed food intake, time on screen, sleeping and physical activity habits.

## 2. Methodology

From 11 to 26 May 2021, parents and legal caregivers reported on lifestyle and sociodemographic variables of children and adolescents from 2 to 18 years old in Chile. Our research question was: How did the COVID-19 pandemic influence the feeding behaviour and lifestyles of children and adolescents one year after lockdown in Chile?

In this cross-sectional study, data were collected using an anonymous and structured online questionnaire delivered by Google Forms. The questionnaire included 8 sociodemographic questions and 22 questions about changes in lifestyle (feeding behaviour, physical activity, on screen time and sleeping habits) during the lockdown caused by the COVID-19 pandemic. The dissemination and distribution of the survey was performed by invitation on social networks (i.e., WhatsApp, LinkedIn, Twitter, Instagram and Facebook) and contacts of the researchers and collaborators. Geographical areas of the country were grouped in 4 macrozones (North, Central, South and Metropolitan Area). In addition, respondents were characterized according to the type of relationship with child or adolescent (mother, father, grandparent or legal caregiver), age group (18–25, 26–35, 36–45 and >46 years), educational level (elementary/high school, undergraduate degree, postgraduate degree), sex (female, male), number of members in the family group (≤3, 4–5, ≥6) and age group of children and adolescents (pre-school: 2–5 years old, school: 6–10 years old and adolescents: 11–18 years old).

Feeding behaviour (FB) was assessed using the intake frequency of dairy products, fruits, vegetables, legumes, fish, liquids, ultra-processed foods (sausages, junk food and snacks) and the choice of shopping foods with Chilean front-of-package nutrition labelling was recorded (i.e., “high in” calories, sugar, saturated fat or sodium). Furthermore, the number and type of daily meals and snacks were also recorded to assess FB according to the recommendations provided by the Food Guidelines for Children (over 2 years of age) and Adolescents proposed in 2015 by the Chilean Ministry of Health [15].

Leisure activities were recorded as daily hours spent on screen, use of screens during feeding and the hours of daily sleeping. All variables were evaluated according to the specific recommendations for pre-schoolers, schoolchildren and adolescents provided by the Food Guidelines for Children (over 2 years of age) and adolescents [15], recommendations for children and adolescents in the Chilean Lockdown Protocol (“Plan paso a paso”, 2020) [16], the new WHO guidelines of 2019 on physical activity, sedentary lifestyle and sleeping habits for children under 5 years old [17] and Guidelines on physical activity and sedentary behaviour of 2020 [18].

Weekly hours of physical activity were categorized according to international guidelines (WHO). Physical activity was declared as “performing” when the respondent declared at least 1 h of physical activity per day [17,18].

Initially, each question was categorized and thoroughly analysed regarding the compliance with national and international recommendations of lifestyles. Compliance to these guidelines was defined for each category as follows: “eat breakfast daily”: always, “daily meals”: breakfast-lunch-afternoon snack-dinner; “type of snacks”: fruits, vegetables, dairy products, nuts, sugar-free cereals and bread, “dairy products intake”: ≥3 servings per day, “fruit intake”: ≥2 servings per day, “vegetable intake”: ≥2 servings per day, “fish intake”: ≥2 times per week, “legumes intake”: ≥2 times a week, “intake of non-dairy liquids”: water, “overnight food intake (between 10 pm and 5 am)”: never, “processed meats (sausages, cold cuts, hamburgers, nuggets) intake”: no intake or less than 1 or 2 times per month, “fast food (hot dogs, hamburgers, french fries, pizza, sushi and others) intake”: no intake or less than 1 or 2 times per month, “sweet snacks (cookies, chocolates or sweets) or salty snacks (potato chips or ultra-processed corn tortillas flavoured with cheese, ham or others) intake”: no intake or less than 1 or 2 times per month, eating while watching screens (TV, mobile phone, computer or notebook, tablet or video game consoles)”: never, “hours on screen per day”: no screens or less than 1 h per day for pre-schoolers and no screens or ≤2 h per day for school children and adolescents, “hours of physical activity”: at least 1 h per day, “sleeping hours per day”: ≥11 h a day for pre-schoolers, 9–12 h a day for schoolchildren, 7–10 h a day for adolescents.

Questions regarding feeding behaviour were grouped and evaluated independently. Subsequently, these questions were added to the lifestyle item. Furthermore, a score based on both items (FB and lifestyle) was created by adding the score obtained from each Likert-type scale question of the survey. Finally, the FB was categorized in three levels: unhealthy (Percentile_<50_; FB <30 points), requires changes (Percentile_50–74_; FB 30–32 points) and healthy (Percentile_≥75_; FB > 32 points). The maximum score was 42 points. Similarly, lifestyle was categorized in three levels: unhealthy (Percentile_<50_; lifestyle score <37 points), requires changes (Percentile_50–74_; lifestyle score 37–39 points) and healthy (Percentile_≥75_; lifestyle score > 39 points). The maximum score was 51 points.

### 2.1. Statistical Analysis

Data are presented using frequencies for categorical variables and mean and standard deviation for continuous variables. The association of sociodemographic factors with FB and lifestyle score was analysed using the Chi-square test and the two-sample proportion test. Significance was set at *p* < 0.05. Statistical analyses were performed using SPSS^®^ (IBM; New York, NY, USA) 21.0.

### 2.2. Ethics

The study was performed in accordance with the Declaration of Helsinki. This study was approved by the Universidad Finis Terrae Bioethics Committee (#13-06-2021). All participants in the study (parents and legal caregivers of children and adolescents) provided a digital signed informed consent before answering the questionnaire.

## 3. Results

One thousand and eighty-three surveys were completed. Sociodemographic and nutritional characteristics of the study group are presented in Table 1. Most of the respondents (49.4%) were from the Metropolitan Area of Chile. Mean and (SD) age of children and adolescents were 9.2 (4.6) years. Most of the children and adolescents were women (51.2%). Mean and (SD) age of parents or caregivers were 38.6 (7.6) years. Regarding the relationship with children and adolescents, 86.9% declared to be the mother. Furthermore, 80.4% of respondents declared that they had an undergraduate or a postgraduate degree.

Table 2 shows that compliance of the recommendations for healthy lifestyles is lower than 70%, except for “eat breakfast every day” and “sleeping hours” in all age groups of children and adolescents. There are significant differences in the compliance of healthy lifestyles between age groups. The compliance was significantly higher in pre-schoolers compared to schoolchildren and adolescents (*p* < 0.05). This was not the case for “food intake while on screen”. Sleeping hours per day was significantly higher in adolescents compared to pre-school and school children.

In all age groups, the higher compliance in FB items were “eat breakfast daily” (89.2%), “overnight food intake” (69.9%) and “no intake of fast food” (66%). Overall, the compliance to recommendations were below 50% for the intake of dairy products (28.6%), fruits (44.8%), vegetables (46.9%), fish (21.2%) and legumes (31.7%). The item “fish intake ≥2 per week” did not show significant differences between groups. The fruit and vegetable intake was higher compared to fish intake and significantly higher in pre-schoolers compared to schoolers and adolescents. Overall, the compliance to guidelines for screen time and physical activity was low. Physical activity was significantly higher in pre-schoolers compared to schoolers and adolescents. Time on screen was significantly higher in schoolers compared to pre-schoolers and adolescents.

Table 3 shows the responses to FB and lifestyle items grouped by sex and age group. Respondents declaring healthy FB and lifestyle were 23.4% and 23.7%, respectively. These proportions were similar in both sexes (*p* > 0.05). In the age group analysis, pre-schoolers reported significantly higher healthy FB and lifestyle compared to school children and adolescents. Table 4 shows responses to FB and lifestyle grouped by the number of family members and educational level of parents and caregivers. Respondents declaring ≤3 family members and parents and caregivers with an undergraduate or a postgraduate degree have significantly healthier FB and lifestyle.

## 4. Discussion

There is little, but growing information regarding the lifestyle of children and adolescents in Chile and world-wide. Most of the studies performed at the beginning or during lockdown due to the COVID-19 pandemic have investigated mainly adolescents and school children [9,19,20,21,22]; but the evidence is still scarce in pre-school children. The purpose of our study was to examine the feeding behaviour and lifestyle of children and adolescents one year after the lockdown by COVID-19 in Chile. We found that eating breakfast every day, eating overnight and eating fast food were the items with higher compliance of international and national recommendations in pre-schoolers. The classification of healthy on EB and lifestyle in our study was of 23.4 and 23.7%, respectively, with no differences by sex. In the subgroup analysis, we found statistically greater compliance of the recommendations in pre-school children, families with fewer than 3 members and parents or caregivers with at least an undergraduate or postgraduate degree.

### 4.1. Pre-School Children

Overall, the FB alone as well as the lifestyle in pre-school children were better and statistically significant compared to school children and adolescents, except in the hours of daily sleep (*p* < 0.05). This may be explained by a high level of dependence on parents and caregivers that pre-school children have and a high pressure on parents and caregivers to achieve healthier FB and lifestyle in children [23]. It is important to highlight that although our results are better compared to other age groups, they did not show a positive compliance to the recommendation, with the exception of the “eat breakfast every day” (93.8%), “overnight food intake” (81.6%) and “eat fast food” (75.6%) items that achieved high compliance in our cohort. Thus, these results should be taken with caution and understanding that it is essential to continue strengthening health policies on healthy lifestyles to prevent of childhood obesity.

A study by Brzęk et al., 2021, in a sample of 1316 pre-schoolers aged 3 to 5 years in Poland, observed a decrease in compliance with WHO recommendations for physical activity (30.8 vs. 21.1%), hours spent sitting (21.9 vs. 9.8%), screen time (26.9 vs. 11.0%) and increased sleeping time (60.9 vs. 74.7%), when comparing the period before and during the COVID-19 pandemic [24]. These results are lower than those observed in our sample, where one year after lockdown, compliance with the recommendations for physical activity, hours of sleep and screen time was 50.3, 39.7 and 16.3%, respectively.

### 4.2. Schoolers and Adolescents

Previous studies showed that during lockdown, school children and adolescents had increased the intake of fresh fruits, juices, vegetables, dairy products, pasta, sweets and snacks and decreased the fast food intake. Furthermore, previous studies reported increased time on screen, sleeping time and reduced physical activity [9,19,20,21,22,25], which is similar to what was observed in our study. Our results suggest that lockdown due to the COVID-19 pandemic, with the concomitant school closing, negatively affected FB and lifestyle of children and adolescents. A study by Ruiz-Roso et al., 2020 in a multinational sample of 820 adolescents from Italy, Spain, Chile, Colombia and Brazil reported that families had more time to cook and improved eating habits by increasing the intake of legumes, vegetables and fruits and reduced fast food consumption during lockdown. However, these improvements were not high enough to increase diet quality because of a higher consumption of sweets and fried foods [9]. Similarly, in the study by Androutsos et al., 2021 at the beginning of the COVID-19 pandemic in 397 schoolchildren and adolescents, the authors observed a significant decrease in the consumption of fast food, which may be explained by fear of being infected by the coronavirus virtually transmitted by delivery drivers [19]. Conversely, Ng et al., 2020 reported that 50% of the adolescents decreased physical activity levels during lockdown in a cohort of 1214 Irish adolescents, which was lower in overweight and obese adolescents [26].

In our study, the declared compliance with the intake of legumes and vegetables in adolescents was 23.7 and 41.2%, respectively. These compliance percentages were similar to the results observed in the study by Ruiz-Roso et al. (24.7 and 43%, respectively). However, the fruit intake was higher in our study when the same intake frequency of once a day was used (51.3 vs. 33.2%, respectively) [9]. However, when using the recommended frequency intake of two or more times a day, the fruit intake percentage dropped to 29.8% in our group of adolescents. The difference with the study by Ruiz-Roso et al. may allow us to establish a certain trend in Chilean adolescents regarding the frequency and intake of legumes and vegetables, which were increased since the beginning of the pandemic. It can be speculated that these results may be explained by more time available to cook at home as well as the increase in legumes and vegetables sales [27], which were the most recommended food by the WHO [28] and the Chilean Ministry of Health [15].

Regarding fast food intake before the pandemic, 44.6% of adolescents reported consuming fast food less than once a week, while once the lockdown started this consumption increased to 64% [9]. However, more than one year after the pandemic, we showed that the percentage of adolescents consuming fast food was 58.8%. This percentage was higher than that previously observed and was similar to what the previous study reported at the beginning of the pandemic [9]. The intake of sweets, fried food and processed meats every day of the week reported prior to the pandemic was 13.4, 2.1 and 6.8%, respectively. These percentages increased to 20.6, 3.2 and 8.4% at the beginning of the lockdown [9]. Interestingly, we found lower intake of sweet/salty snacks and processed meats (8.6% and 2.8%, respectively) in our study using the same intake frequency of previous study [9]. However, the compliance of the present study showed that the recommendation for sweet/salty snacks intake was 24.2% and processed meats was 49% but with much stricter criteria. The FB of eating while looking at screens in adolescents reported by previous study [9] compared with our results was distributed as follows: always: 46.7 vs. 30.8%; sometimes: 19.9 vs. 41.4% and never: 33.4 vs. 27.8%, respectively. Hence, we found a decrease in those who always consumed food looking at screens and an increase in those who did it sometimes. These results are somewhat encouraging compared to those observed prior to confinement and could have a positive impact on the decrease in energy intake, risk of obesity [29,30] and chronic diseases [31]. However, it is complex to assess this potential impact compared to the negative effects that the pandemic may have had on children and adolescents [32,33]. Finally, it should be noted that there are similarities with the Ruiz-Roso et al., 2020 study reported at the beginning of the pandemic [9] and ours. The undergraduate and postgraduate degree educational level of parents or caregivers was predominant (72.4 vs. 80.4%, respectively) in both studies and could be an important factor explaining the positive results observed in some survey items.

Lifestyle modifications in children and adolescents may have different explanations. Changes in eating behaviour may be explained by factors such as insecurity caused by COVID-19 and changes in the household food environment and feeding practices. For example, Adams et al., 2020 reported that families experiencing food insecurity exerted greater pressure on their children to eat, while 30% of families increased the number of high-calorie snacks, desserts, sweets and fresh foods, and almost half of the participants increased availability of non-perishable processed foods in their homes [23]. In addition, changes in the social environment, such as job loss or greater workload of parents, may have generated greater stress in parents, modifying family interactions and feeding practices [19]. In addition, the psychological impact of lockdown and social isolation may have triggered boredom and stress, which are associated with the consumption of more energy-dense foods and emotional eating [34]. Similarly, the study by Pietrobelli et al., 2020 observed that cooking and preparing new recipes with children and adolescents can be used by families as a recreational activity during lockdown, which in turn increases the availability of sweets, snacks and home-made foods [12].

The increase in screen time may be explained by the implementation of virtual classes due to the closure of kindergartens, public and private schools, as well as more free time at home and boredom. The increased duration of sleep, especially in school children and adolescents, could be related to the fact that attending school in the morning was no longer necessary. The decrease in physical activity levels can be attributed to home lockdown, which does not allow people to attend sports clubs and structured exercise activities or visiting schoolyards, parks and recreational areas [19,26]. Our results revealed that only 25.9% of the children and adolescents included in our study followed international recommendations regarding the practice and frequency of physical activity. These results are consistent with other studies at the national level and reveal a worrying trend to sedentarism in these age groups. Furthermore, it is important to highlight that physical activity habits also define an important part of lifestyle and therefore correspond to a fundamental dimension of quality of life in people [35,36]. The WHO recommend that children and young people engage in at least 60 min of physical activity daily [18]. It is known that at these stages of life, the constant practice of games, trips, outdoor activities, exercises and sports practice should be favoured during leisure time, as well as during physical education classes [37]. Our results also indicate that school children and adolescents have less physically active lifestyles than pre-schoolers. The National Survey of Physical Activity and Sports Habits developed by the Sports Ministry of Chile in 2019 reported similar results. This survey included 5059 Chilean children and adolescents and revealed that the percentage of active children (defined as those who performed physical and/or sports activity for a minimum of 60 min daily) decreased with age. In our study, approximately 19% of children between 5 and 12 years old were categorized as active, while only 10.8% of adolescents between 13 and 17 years old complied with the recommendations for young people [35]. It can be seen from these results that age is a determinant factor not only for feeding behaviour but also for those with healthy habits in terms of physical activity. As unhealthy lifestyles can lead to obesity, metabolic complications and increase the severity of COVID-19 symptoms, it is then essential to promote healthy feeding, sleep hygiene and physical activity during lockdown [6,12,13,38,39].

Of the strengths in our study, the approach to feeding behaviour and lifestyles stands out, as well as providing information that can contribute to measuring the impact of the COVID-19 pandemic on children and adolescents in Chile.

Our study is not without limitations. Given that this is a cross-sectional study, a cause-and-effect relationship cannot be established. Although the sample size is big, it cannot be assumed to be representative of all children and adolescents in Chile. Therefore, the results cannot be extrapolated to the entire population. It is also assumed that the results of the survey may be biased due to the high educational level of the respondents of the study, as well as the natural tendency to respond to what is socially desirable.

## 5. Conclusions

A significantly higher proportion of pre-school children showed a healthy feeding behaviour and healthy lifestyle during lockdown one year after the COVID-19 pandemic compared to school children and adolescents. Sociodemographic factors such as the number of members in the family group and educational level of parents and caregivers were also significantly associated with greater compliance with national and international lifestyles recommendations. These results are of concern and reflect that lockdown can have deleterious consequences at personal, social, psychological and family level, which may produce imbalances in the maintenance and acquisition of healthy lifestyles behaviours. National and international studies that have addressed aspects of lifestyle in childhood and adolescent stages are scarce. Therefore, these results contribute to better identifying the impact and repercussions that the COVID-19 pandemic has had on children and adolescents in Chile.

## Figures and Tables

**Table 1 nutrients-13-04138-t001:** Sociodemographic and nutritional characteristics of the study groups.

Variables	% of Sample	Variables	% of Sample
Geographic zone		Age group of the child or adolescent (years)	
North	6.0	Pre-schooler (2–5)	29.5
Centre	17.8	Scholar (6–10)	33.9
Metropolitan Area (MA)	49.4	Adolescent (11–18)	36.6
South	26.8	In pandemic: Did you check the “High In” warning stamps to buy groceries and food?	
Relationship		Always	32.5
Mother	86.9	Sometimes	47.0
Father	10.2	We used to do it. Not now	4.2
Grandfather/mother	1.3	Never	16.3
Legal caregiver	1.6	How much your family ate during the COVID-19 pandemic compared to before	
Age group of parents or caregivers (years)		Much more than before	15.1
18 to 25	3.4	Much less than before	1.1
26 to 35	30.5	Same	31.1
36 to 45	47.2	A little bit more than before	46.2
>46	18.9	A little bit less than before	6.6
Educational level		Main ways your home meals were prepared	
Elementary/High school	19.6	Home-made meals with raw and natural food	86.6
Undergraduate	60.8	Home-made meals with processed food	12.7
Postgraduate	19.6	Restaurant or delivery meals	0.4
Number of members in the family group		Supermarket prepared meals	0.3
≤3	33.0	Your child participated in preparing meals at home	
4–5	58.9	Always	3.4
≥6	8.1	Sometimes	64.4
Sex of the child or adolescent		Never	32.2
Female	51.2		
Male	48.8		

**Table 2 nutrients-13-04138-t002:** Percentage of compliance with the Healthy Lifestyles recommendations for children and adolescents.

	Total (*n* = 1083)		Pre-Schoolers (*n* = 320)		Schoolers (*n* = 367)		Adolescents (*n* = 369)		*p*-Value
	*n*	%	*n*	%	*n*	%	*n*	%	
Feeding behaviour									
Eat breakfast daily	996	89.2	300	93.8 ^a^	349	95.1 ^a^	317	80.1 ^b^	<0.001 *
Daily meals (breakfast-lunch-tea-time-dinner)	374	34.5	169	52.8 ^a^	108	29.4 ^b^	97	24.5 ^b^	<0.001 *
Type of snacks	605	55.9	191	59.7	193	52.6	221	55.8	0.174
Dairy intake (≥3 per day)	310	28.6	146	45.6 ^a^	106	28.9 ^b^	58	14.6 ^c^	<0.001 *
Fruit intake (≥2 per day)	485	44.8	204	63.7 ^a^	163	44.4 ^b^	118	29.8 ^c^	<0.001 *
Vegetable intake (≥2 per day)	508	46.9	178	55.6 ^a^	167	45.5 ^b^	163	41.2 ^b^	<0.001 *
Fish intake (≥2 week)	230	21.2	74	23.1	82	22.3	74	18.7	0.288
Legume intake (≥2 week)	343	31.7	133	41.6 ^a^	116	31.6 ^b^	94	23.7 ^c^	<0.001 *
Liquid drink (no dairy)	276	25.5	119	37.2 ^a^	82	22.3 ^b^	75	18.9 ^b^	<0.001 *
Overnight food intake	757	69.9	261	81.6 ^a^	268	73.0 ^b^	228	57.6 ^c^	<0.001 *
Processed meats intake	539	49.8	193	60.3 ^a^	152	41.4 ^b^	194	49.0 ^c^	<0.001 *
Fast food intake	715	66.0	242	75.6 ^a^	240	65.4 ^b^	233	58.8 ^b^	<0.001 *
Sweet or salty snacks intake	250	23.1	87	27.2 ^a^	67	18.3 ^b^	96	24.2 ^a^	0.017 *
Food intake while looking at screens	269	24.8	65	20.3	94	25.6	110	27.8	0.065
Leisure, physical activity and sleeping habits									
Hours of screens per day	161	15.8	52	16.3 ^a^	87	23.7 ^b^	32	8.1 ^c^	<0.001 *
Hours of physical activity	281	25.9	161	50.3 ^a^	66	18.0 ^b^	54	13.6 ^b^	<0.001 *
Hours of sleep per day	770	71.1	127	39.7 ^a^	280	76.3 ^b^	363	91.7 ^c^	<0.001 *

Chi-squared tests. * *p* < 0.05; ^a,b,c^ Two-sample proportion test. *p* < 0.05.

**Table 3 nutrients-13-04138-t003:** Feeding behaviour and lifestyle by sex and age range of children and adolescents.

	Total (*n* = 1083)		Female (*n =* 554)		Male (*n =* 529)		*p*-Value	Pre-School Children (*n* = 320)		School Children (*n* = 367)		Adolescent (*n* = 369)		*p*-Value
	*n*	%	*n*	%	*n*	%		*n*	%	*n*	%	*n*	%	
Feeding behaviour														
Unhealthy	447	41.3	214	38.6	233	44.0	0.162	81	25.3	171	46.6	195	49.2	<0.001 *
Requires changes	383	35.4	201	36.3	182	34.4		107	33.4	128	34.9	148	37.4	
Healthy	253	23.4	139	25.1	114	21.6		132	41.3 ^a^	68	18.5 ^b^	53	13.4 ^b^	
Lifestyles														
Unhealthy	518	47.8	259	46.8	259	49.0	0.395	107	33.4	171	46.6	240	60.6	<0.001 *
Requires changes	308	28.4	154	27.8	154	29.1		86	26.9	117	31.9	105	26.5	
Healthy	257	23.7	141	25.5	116	21.9		127	39.7 ^a^	79	21.5 ^b^	51	12.9 ^c^	

Chi-squared tests. * *p* < 0.05; ^a,b,c^ Two-sample proportion test. *p* < 0.05.

**Table 4 nutrients-13-04138-t004:** Feeding behaviour and lifestyles of children and adolescents according to the number of members in the family group and educational level of parents and caregivers.

	≤3 Members (*n* = 357)		4 to 5 Members (*n* = 638)		≥6 Members (*n* = 88)		*p*-Value	Elementary/High School (*n* = 212)		Undergraduate (*n* = 659)		Postgraduate (*n* = 212)		*p*-Value
	*n*	%	*n*	%	*n*	%		*n*	%	*n*	%	*n*	%	
Feeding behaviour														
Unhealthy	147	41.2	274	42.9	26	29.5	<0.001 *	106	50.0	276	41.9	65	30.7	<0.001 *
Requires changes	103	28.9	235	36.8	45	51.1		75	35.4	229	34.7	79	37.3	
Healthy	107	30.0 ^a^	129	20.2 ^b^	17	19.3 ^c^		31	14.6 ^a^	154	23.4 ^b^	68	32.1 ^c^	
Lifestyles														
Unhealthy	173	48.5	309	48.4	36	40.9	0.004 *	120	56.6	321	48.7	77	36.3	<0.001 *
Requires changes	80	22.4	198	31.0	30	34.1		56	26.4	189	28.7	63	29.7	
Healthy	104	29.1 ^a^	131	20.5 ^b^	22	25.0 ^a^^,^^b^		36	17.0 ^a^	149	22.6 ^a^	72	34.0 ^b^	

Chi-squared tests. * *p* < 0.05; ^a,b,c^ Two-sample proportion test. *p* < 0.05.

## Data Availability

All data analysed in this study are available upon request to the corresponding author.

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
