# Peer review of "Feeding Behaviour and Lifestyle of Children and Adolescents One Year after Lockdown by the COVID-19 Pandemic in Chile"

_nutrients, 2021, doi:10.3390/nu13114138_

Round 1

Reviewer 1 Report

Thank you for the opportunity to read the article.
A couple of comments. 
Please add either research questions or hypotheses in the methodology section.
Please mark the significance of p-value with * and not in bold.
Add strengths and limitations at the end of the discussion section.

Remove unnecessary blank pages in the article.

Reviewer 2 Report

I would be interested in the basic data: Do you have information about the weight of the children before and after the pandemic? One would have to know the size and weight of the children in the questionnaire. Please cite and discuss the paper of Brzek et al: 

Brzęk, A., M. Strauss, F. Sanchis-Gomar and R. Leischik (2021). "Physical Activity, Screen Time, Sedentary and Sleeping Habits of Polish Preschoolers during the COVID-19 Pandemic and WHO’s Recommendations: An Observational Cohort Study." International Journal of Environmental Research and Public Health 18(21): 11173.
